# Intrauterine exposure to perfluorinated compounds and overweight at age 4: A case-control study

**Matilda Martinsson**[ID]**, Christel Nielsen**[ID]***, Jonas Björk, Lars Rylander, Ebba Malmqvist, Christian Lindh, Anna Rignell-Hydbom**

Division of Occupational and Environmental Medicine, Lund University, Lund, Sweden

* christel.nielsen@med.lu.se

## Abstract

### Aims

The aims were to investigate the association between maternal serum levels of perfluorooctane sulfonate (PFOS), perfluorooctanoic acid (PFOA), perfluorohexane sulfonate (PFHxS) and perfluorononanoic acid (PFNA) in early pregnancy and overweight in the child at 4 years and to assess potential heterogeneity in exposure effect between strata with different levels of other risk factors for overweight.

### Methods

We used a case-control design and included 354 cases (ISO-BMI $\geq$ 18 kg/m$^2$) and 2 controls per case (ISO-BMI $\leq$17 kg/m$^2$) from child health care centers in Malmö, Sweden. Controls were selected stratified on risk scores for overweight in a propensity score framework. Maternal serum levels were analyzed in biobanked samples collected by routine around gestational week 14. Logistic regression was used to estimate odds ratios between quartiles of maternal serum levels and child overweight at age 4.

### Results

There were no consistent monotonic exposure-response relationships. We found some significant odds ratios in specific quartiles but these were regarded as spurious findings. The absence of an effect was consistent over risk strata.

### Conclusions

We did not find evidence of an association between maternal serum levels of PFOS, PFOA, PFHxS and PFNA in early pregnancy and child overweight at age 4. The level of other risk factors for overweight did not affect children's susceptibility to prenatal PFAS exposure.

**Data Availability Statement:** Data contains sensitive personal information and cannot be shared publicly because of ethical and legal constrictions under GDPR. Before consenting to

participate in the study, participants were informed that the collected data would only be available to the researchers working on the project. This restriction on sharing data was part of the documents reviewed and approved by the Ethical Review Board. A new application to the Swedish Ethical Review Authority is needed if external researchers would like to access the data. Information on how to apply is found at https://etikprovningsmyndigheten.se. Requests for data should be directed to the Head of the Department of Laboratory Medicine, Lund University, at prefekt.ilm@med.lu.se.

**Funding:** This work was supported by grants from the Swedish Research Council for Health, Working Life and Welfare [grant number 2013-001158], The Swedish Research Council, and the Faculty of Medicine, Lund University. The funders had no role in study design, data collection and analysis, decision to publish, or preparation of the manuscript.

**Competing interests:** The authors have declared that no competing interests exist.

# Introduction

Per- and polyfluorinated alkyl substances (PFAS) are man-made compounds and many are persistent in the environment and human body. PFAS are used both in industrial and consumer products, such as food packaging, non-stick pans, waterproof clothing and furniture because of their water- and oil-repelling properties. The most important pathway of exposure for the general population is through food and drinking water [1].

Perfluorooctane sulfonate (PFOS), perfluorooctanoic acid (PFOA), perfluorohexane sulfonate (PFHxS) and perfluorononanoic acid (PFNA) all have biological half-lives of several years [2, 3]. PFAS are transferred from mothers to their children as PFAS cross the placental barrier [4] and the levels in maternal blood samples correlate with the levels in cord blood at birth [5]. PFAS are endocrine disruptors that might influence childhood weight by interfering with endogenous hormonal processes important to program or maintain growth and development. Overweight and obese children suffer an increased risk of staying overweight as adults [6] and child and adolescence overweight and obesity have been associated to premature death [7, 8].

Animal studies show that mice exposed to PFAS during gestation are more likely to become overweight compared to non-exposed mice [9, 10]. Epidemiological studies show associations between prenatal exposure to PFAS and adiposity, body mass index (BMI) [11–13], anthropometry [14], skinfold thickness and fat mass [15] and waist-to-height ratio [16] in children. However, the results are inconsistent and some studies found no associations [17, 18]. Previous studies have limitations that make extrapolation of their findings difficult, e.g. relying on self-reported body measurements and lack of a standardized strategy for sampling of maternal blood.

It has been suggested that the effect of environmental exposures may vary depending on the presence or absence of other risk factors that can be taken to reflect an unobserved variable for susceptibility [19]. Overweight is indeed a multifactorial disorder and it may be hypothesized that the level of other risk factors might render children more or less susceptible to the effect of prenatal PFAS exposure.

The primary aim of the study was to investigate the association between maternal serum levels of PFOS, PFOA, PFHxS and PFNA in early pregnancy and overweight in the child at 4 years of age using data from child health care centers and biobanked serum samples from around gestational week 14. A second aim was to explore potential heterogeneity of exposure effects between children with different levels of other risk factors for overweight.

# Materials & methods

## Study population

Children who underwent routine 4-year health examinations at child health care centers in Malmö, Sweden, during 2003–2008 and whose parents answered a self-administered questionnaire were eligible for inclusion. Child health care centers have scheduled visits with children up to school age and the coverage is close to 100%. The pediatric nurse recruited participants in connection with the visit and measured the weight and height of the child in a standardized manner. The questionnaire, described in more detail by Mangrio et al. [20], included 32 questions related to the family situation, the health of the child, the parents' socioeconomic status, smoking in the family, breastfeeding and intake of soft drinks. The response rate was 68% (*n* = 9009). A flowchart describing the study population is shown in Fig 1.

The study was restricted to children born ≥38 gestation weeks with a birth weight of 2000–6000 g and a weight and height at 4 years of 10–50 kg and 75–125 cm, respectively and with parents born in Sweden. The latter excluded a fairly large proportion of the children as 48% of

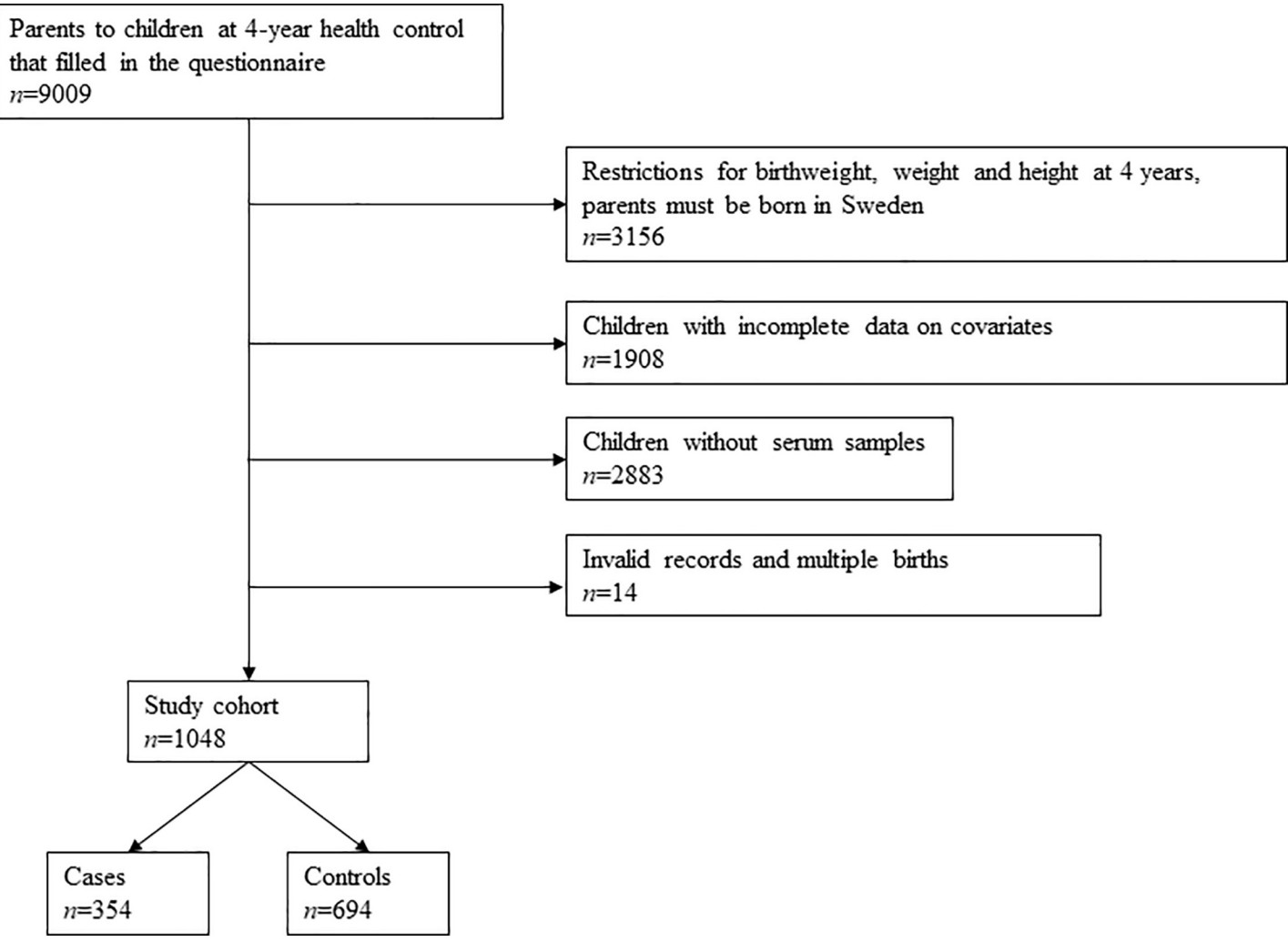

**Fig 1. Flowchart describing the inclusion of the study population.**

children aged 1–5 years in Malmö had another native language than Swedish in 2008 [21]. After these restrictions the study population comprised 5853 children.

The study protocol followed the requirements of the Declaration of Helsinki and was approved by the Regional Ethical Review Board in Lund (2014/696) with waived requirement for informed consent. Data was pseudonymized before the study was conducted.

## Participants

**Cases and controls.**   Age and sex adjusted body mass index (ISO-BMI) was calculated. Cases ($n$ = 545) were defined as having ISO-BMI$\geq$18 kg/m$^2$, corresponding to a BMI of 25 kg/m$^2$ in adults [22], and children with ISO-BMI$\leq$17 kg/m$^2$ ($n$ = 4472) were considered controls. Children with 17<ISO-BMI<18 (n = 836) were excluded to create contrast between cases and controls.

**Control-selection strategy.**   Simple random selection of controls in a case-control study is often inefficient from a statistical perspective if there are known risk factors for the outcome [19]. Matching is traditionally performed to increase efficiency under the presence of one or

few risk factors but it often becomes impractical in the case of a multifactorial disorder such as overweight. Therefore, we applied a two-step control-selection strategy based on risk scores in a propensity score framework that has been shown to improve statistical precision. The methodological approach is described in detail by Björk et al. [19].

Firstly, the risk (i.e. the propensity) of being overweight was modelled using logistic regression based on data on from the questionnaire. The risk model included maternal smoking during pregnancy (0/1), birth weight (continuous), economic strain (defined as not being able to buy adequate amounts of food and clothing for more than 6 months during the last year; 0/1), being a tenant (0/1), maternal obesity (0/1; BMI$\geq$30 kg/m$^2$) and paternal obesity (0/1; BMI$\geq$30 kg/m$^2$). The child's order among the siblings, sex, parental education and the child's intake of soft drinks were considered as covariates but were not retained in the final risk model because of $p$-values >0.05 and an OR$\leq$1.5 [19].

Secondly, we used the risk model to estimate risk scores for all children with complete data on the covariates ($n$ = 3945) and categorized them into three risk strata: low (0–5%; $n$ = 950), intermediate (6–13%; $n$ = 2154) and high ($\geq$14%; $n$ = 841) risk. The cut-offs were set using a data-driven approach with a factor 2 increase in risk between the strata to create sufficient contrast between the low and the intermediate strata. All cases ($n$ = 416) were retained for the final analyses and matched with 2 controls per case ($n$ = 832) from the same risk strata, with further matching on sex.

## Exposure assessment

The Southern Sweden Maternity Cohort is a population-based biobank containing >250,000 samples collected at the routine serological screening for viral infections and rubella immunity around gestational week 14. Pregnant women were asked by their maternal health care clinic whether their samples could be used for future purposes in an opt-out manner. This is standard procedure for all biobanks within Swedish health care. Serum samples from the mothers of 1062 study participants (85%) were available.

**Chemical analyses.** Serum concentrations of PFOS, PFOA, PFHxS and PFNA were analyzed at the Division of Occupational and Environmental Medicine in Lund, Sweden, using liquid chromatography-tandem-mass-spectrometry (LC/MS/MS) and the method previously described by Lindh et al. [23] with minor modifications. In brief, aliquots of 75 µl serum were added with glucuronidase, ammonium acetate buffer and labeled internal standards and digested at 37˚C for 90 min. The proteins were then precipitated with 200 µl acetonitrile and vigorously shaken for 30 min. The samples were centrifuged and 4 µl of the supernatant was analyzed using LC/MS/MS (QTRAP5500, Applied Biosystems, Foster City, CA, USA). Each analytical batch included calibration standards, two quality control samples (QC) and chemical blank samples.

The samples were analyzed in duplicate and in a randomized order. The limit of detection (LOD) was 0.07 ng/ml for PFOS, 0.04 ng/ml for PFOA and 0.03 ng/ml for PFHxS and PFNA. Coefficient of variation (CV) of the QC samples (n = 32) was 7% at 12 ng/ml and 8% at 13ng/ml for PFOS. For PFOA, the CV was 12% at 3 ng/ml and 9% at 4 ng/ml, for PFHxS the CV was 8% at 2 ng/ml and 10% at 3 ng/ml, and for PFNA the CV was 10% at 2 ng/ml and 9% at 4 ng/ml. The analyses of PFOS and PFOA are part of a quality control program between analytical laboratories coordinated by Professor Hans Drexler, Institute and Outpatient Clinic for Occupational, Social and Environmental Medicine, University of Erlangen-Nuremberg, Germany. All samples had concentrations above LOD for the analyzed PFAS.

## Statistical analyses

The final dataset contained 1048 children (cases n = 354, controls n = 694) after data cleansing (i.e. exclusion of invalid records and restriction to singleton births). We did not restrict the

number of children per mother because we wanted to maximize the number of cases eligible for inclusion. Consequently, 4% of the women had multiple children in the dataset and this introduced a hierarchical data structure. Multilevel logistic regression with a random intercept for each mother enabled retention of all cases in the analyses without violating the assumption of independence. The SAS/STAT (version 9.4 for Microsoft Windows; SAS Institute Inc., Cary, NC) procedure GLIMMIX with a logit link function was used to model the associations between overweight (cases: ISO-BMI$\geq$18 kg/m$^2$; controls: ISO-BMI$\leq$17 kg/m$^2$) and quartiles of PFOS, PFOA, PFHxS and PFNA defined according to the distributions among the controls. Analyses were performed separately for each PFAS (single-exposure models) as well as jointly for all PFAS (multi-exposure model; Spearman correlations between different PFAS after quartile split were in the range of 0.24 to 0.60). Models were run 1) on the *full dataset*, adjusting for risk strata, difference from the strata-specific mean and sex, and 2) *stratified on risk strata*, adjusting for difference from the strata-specific mean and sex. We explored interactions between exposures and risk strata in the full-dataset analyses but the terms were non-significant and therefore not included in the final models. *P*-values $\leq$0.05 were considered statistically significant.

## Results

Stratification by risk made cases and controls similar with respect to the covariates in the risk model (Table 1). The median maternal serum concentrations (Q1;Q3) were 16.6 ng/mL (12.6;22.0) for PFOS, 3.1 ng/mL (2.4;4.2) for PFOA, 0.7 ng/mL (0.5;1.0) for PFHxS and 0.4 ng/mL (0.3;0.5) for PFNA. Maternal serum concentrations of PFAS were similar over the risk strata, as shown in Table 2.

There were no monotonic exposure-response relationships in the analyses of the full dataset (Table 3). PFHxS had a significant overall association with the outcome (*p* = 0.02 in the multi-exposure model) and there were some significant odds ratios in specific quartiles for PFOS, PFHxS and PFNA, but with no patterns over the quartiles. The stratified analyses showed that the absence of an effect of maternal PFAS concentrations in early pregnancy on child overweight was consistent over the risk strata (Table 4).

## Discussion

We did not find any monotonic exposure-response relationships between prenatal exposure to PFOS, PFOA, PFHxS and PFNA and overweight in 4-year-old children from the general population, nor did we find evidence of heterogeneous exposure effects over risk strata. Although some significant odds ratios were observed between different quartiles of PFOS, PFHxS and PFNA, no consistent patterns emerged and we regarded these findings as spurious.

Previous studies report inconsistent results regarding the effect of PFAS on overweight in children. Prenatal exposure to PFOS has been associated with increased weight in girls at 20 months [24]. PFOS and PFOA combined were associated with increased waist-to-height ratio in 5 to 9-years-old girls [16], whereas another study [12] found a tendency to positive associations between PFOS and weight in children from age 3 months to 5 years and BMI in age 4 to 5 years. Lauritzen et al. [13] found associations between both PFOS and PFOA and BMI, triceps skinfolds and child overweight/obesity in 5-year-old children. However, other studies have reported no association between PFOS and weight and body measures in children [11, 25].

Associations have been reported between prenatal exposure to PFOA and greater adiposity and increased BMI in 8-year-old children [11] and higher BMI, skin-fold thickness and total fat mass in 6 to 10-year-old children [15]. On the opposite, Shoaff et al. [14] have reported an

**Table 1. Demographic details of the study participants and distribution of the covariates in the risk model for overweight at 4 years of age for cases (ISO-BMI $\geq$ 18 kg/m$^2$) and controls (ISO-BMI $\leq$ 17 kg/m$^2$), stratified by risk.**

| | Total | | | | Low risk | | | | Intermediate risk | | | | High risk | | | |
| --- | --- | --- | --- | --- | --- | --- | --- | --- | --- | --- | --- | --- | --- | --- | --- | --- |
| | Cases [a] (n = 354) | | Controls [b] (n = 694) | | Cases (n = 42) | | Controls [b] (n = 84) | | Cases (n = 159) | | Controls (n = 327) | | Cases (n = 153) | | Controls (n = 283) | |
| Risk factor | % | Median (Q1;Q3) | % | Median (Q1;Q3) | % | Median (Q1;Q3) | % | Median (Q1;Q3) | % | Median (Q1;Q3) | % | Median (Q1;Q3) | % | Median (Q1;Q3) | % | Median (Q1;Q3) |
| Maternal obesity [c] | 14.1 | | 10.1 | | 0 | | 0 | | 1.9 | | 0.3 | | 30.7 | | 24.4 | |
| Paternal obesity [d] | 15.0 | | 11.5 | | 2.4 | | 0 | | 8.2 | | 5.5 | | 25.5 | | 21.9 | |
| Smoking during pregnancy | 11.9 | | 10.5 | | 0 | | 0 | | 8.2 | | 4.3 | | 19.0 | | 20.9 | |
| Economic strain [e] | 4.2 | | 3.3 | | 0 | | 0 | | 2.5 | | 0.6 | | 7.2 | | 7.4 | |
| Tenant | 22.3 | | 18.6 | | 4.8 | | 1.2 | | 17.6 | | 14.4 | | 32 | | 28.6 | |
| Birth weight, g | | 3750 (3460; 4105) | | 3725 (3395; 4060) | | 3410 (2980; 3570) | | 3365 (3205; 3553) | | 3700 (3410; 3980) | | 3675 (3360; 3980) | | 4015 (3695; 4370) | | 3920 (3650; 4295) |
| Sex (girls) | 50.0 | | 50.9 | | 61.9 | | 63.1 | | 52.8 | | 52.0 | | 43.8 | | 45.9 | |

[a] ISO-BMI $\geq$ 18

[b] ISO-BMI $\leq$ 17

[c,d] BMI $\geq$ 30 kg/m$^2$

[e] Defined as being unable to buy adequate amounts of food and clothes for more than 6 months during the last year.

inverse association between PFOA and child anthropometry in children aged from 4 weeks to 2 years.

Lee et al. [26] reported postnatal levels of PFHxS to be associated with a smaller weight gain from birth to 2 years of age. Other studies have found associations between PFHxS and increased adiposity [15] and BMI [12]. Yet others reported no associations between PFHxS and weight, BMI and adiposity in children [11, 27, 28].

Lee et al. [26] reported an inverse association between weight in 2-year old children and postnatal PFNA exposure levels but Gyllenhammar et al. [12] showed a positive association between PFNA and BMI in 3 to 4-year-old children. Mora et al. [15] used exposure samples

**Table 2. Maternal serum concentrations (median (Q1;Q3)) of PFOS, PFOA, PFHxS and PFNA (ng/mL) in early pregnancy and the child's ISO-BMI at age 4, stratified by risk.**

| | Low risk | | Intermediate risk | | High risk | |
| --- | --- | --- | --- | --- | --- | --- |
| Variable | Cases [a] (n = 42) | Controls [b] (n = 84) | Cases (n = 159) | Controls (n = 327) | Cases (n = 153) | Controls (n = 283) |
| PFOS | 18.8 (12.2;26.8) | 16.6 (12.0;20.2) | 17.2 (13.5;24.2) | 16.9 (12.1;22.1) | 16.4 (13.4;21.7) | 15.7 (12.3;21.3) |
| PFOA | 3.3 (2.9;4.6) | 3.2 (2.4;4.4) | 3.1 (2.4;4.3) | 3.2 (2.3;4.3) | 3.2 (2.5;4.2) | 3.0 (2.3;4.0) |
| PFHxS | 0.7 (0.5;1.1) | 0.8 (0.5;1.0) | 0.7 (0.5;1.0) | 0.7 (0.5;0.9) | 0.6 (0.5;1.0) | 0.7 (0.5;0.9) |
| PFNA | 0.4 (0.3;0.6) | 0.4 (0.3;0.6) | 0.4 (0.3;0.5) | 0.4 (0.3;0.6) | 0.4 (0.3;0.5) | 0.4 (0.3;0.5) |
| ISO-BMI | 18.5 (18.2;18.9) | 15.5 (14.8;16.2) | 18.5 (18.2–19.2) | 15.7 (15.0;16.2) | 18.8 (18.3;19.6) | 15.8 (15.1;16.3) |

PFOS: perfluorooctane sulfonate; PFOA: perfluorooctanoic acid; PFHxS: perfluorohexane sulfonate; PFNA: perfluorononanoic acid.

[a] ISO-BMI $\geq$ 18

[b] ISO-BMI $\leq$ 17

**Table 3. Single- and multi-exposure models regressing child overweight at 4 years on quartiles of maternal serum concentrations of PFOS, PFOA, PFHxS and PFNA in early pregnancy.** The analyses were adjusted for risk strata, difference from the strata-specific mean and sex and the quartiles were based on the distribution among the controls.

| Compound (ng/mL) | n | Single-exposure model | | Multi exposure models | |
| | | Odds ratio (95% CI) | p-value | OR (95% CI) | p-value |
| --- | --- | --- | --- | --- | --- |
| PFOS | | | 0.13 | | 0.10 |
| 2.93–12.24 | 239 | 1.00 | | 1.00 | |
| 12.25–16.34 | 272 | 1.42 (0.96–2.09) | | 1.53 (1.01–2.31) | |
| 16.35–21.61 | 261 | 1.26 (0.85–1.87) | | 1.41 (0.90–2.20) | |
| 21.62–65.67 | 276 | 1.57 (1.07–2.30) | | 1.82 (1.11–3.00) | |
| PFOA | | | 0.69 | | 0.92 |
| 0.88–2.32 | 247 | 1.00 | | 1.00 | |
| 2.33–3.09 | 269 | 1.12 (0.76–1.64) | | 1.09 (0.73–1.64) | |
| 3.10–4.19 | 266 | 1.23 (0.84–1.79) | | 1.15 (0.74–1.79) | |
| 4.20–15.07 | 266 | 1.22 (0.84–1.79) | | 1.04 (0.64–1.71) | |
| PFHxS | | | 0.04 | | 0.02 |
| 0.11–0.50 | 271 | 1.00 | | 1.00 | |
| 0.51–0.68 | 264 | 0.95 (0.66–1.37) | | 0.93 (0.63–1.38) | |
| 0.69–0.93 | 233 | 0.66 (0.44–0.97) | | 0.62 (0.40–0.96) | |
| 0.94–24.92 | 280 | 1.16 (0.81–1.66) | | 1.16 (0.75–1.80) | |
| PFNA | | | 0.20 | | 0.08 |
| 0.08–0.30 | 281 | 1.00 | | 1.00 | |
| 0.31–0.39 | 264 | 0.74 (0.52–1.06) | | 0.71 (0.49–1.04) | |
| 0.40–0.52 | 255 | 0.75 (0.52–1.08) | | 0.66 (0.44–0.98) | |
| 0.53–1.53 | 248 | 0.69 (0.48–1.01) | | 0.59 (0.39–0.91) | |

PFOS: perfluorooctane sulfonate; PFOA: perfluorooctanoic acid; PFHxS: perfluorohexane sulfonate; PFNA: perfluorononanoic acid.

from early pregnancy reported PFNA to be associated with increased adiposity in girls in mid-childhood.

It may be that the levels of PFAS in the general population in our study were not high enough to trigger the complex hormonal systems that might affect child weight. However, the serum levels of PFOS and PFOA in this study were well in line with those of other studies, although the levels of PFHxS and PFNA were in the lower range. The difference in maternal PFAS concentrations over risk strata was negligible thus suggesting that the variables in the risk model, all related to socioeconomic status, were not associated with maternal serum concentrations.

It is difficult to make comparisons with the results of previous studies as exposure assessment has occurred at different time periods and different outcomes have been investigated in children of varying ages. PFAS concentrations measured in different trimesters, or before and after delivery, are not comparable because PFAS concentrations change as a result of pregnancy-induced hemodynamic changes and alterations of the glomerular filtration rate [29].

However, the potential impact of publication bias must be acknowledged. In a review of systematic reviews and meta-analyses of environmental exposures and adverse health outcomes, Sheehan and Lam [30] found that approximately half of the reviews that investigated publication bias found evidence of it. As such, there may be a favorable selection of positive associations in the literature.

We speculated that children would be more or less vulnerable to the effect of prenatal PFAS exposure depending on the level of other risk factors present. We did however not find any

**Table 4. Multi-exposure models regressing child overweight at 4 years on quartiles of maternal serum concentrations of PFOS, PFOA, PFHxS and PFNA in early pregnancy, stratified by risk of overweight according to other risk factors.** The analyses were adjusted for risk strata, difference from the strata-specific mean and sex and the quartiles were based on the distribution among the controls.

| Compound (ng/mL) | Low risk | | | Intermediate risk | | | High risk | | |
|---|---|---|---|---|---|---|---|---|---|
| | n | OR (95% CI) | p-value | n | OR (95% CI) | p-value | n | OR (95% CI) | p-value |
| PFOS | | | 0.19 | | | 0.28 | | | 0.35 |
| 2.93–12.24 | 33 | 1.00 | | 114 | 1.00 | | 92 | 1.00 | |
| 12.25–16.34 | 22 | 0.60 (0.14–2.64) | | 118 | 1.60 (0.85–3.02) | | 132 | 1.84 (0.95–3.56) | |
| 16.35–21.61 | 41 | 1.33 (0.34–5.31) | | 115 | 1.47 (0.75–2.88) | | 105 | 1.62 (0.77–3.37) | |
| 21.62–65.67 | 30 | 3.59 (0.70–18.33) | | 139 | 2.04 (0.98–4.26) | | 107 | 1.58 (0.68–3.65) | |
| PFOA | | | 0.72 | | | 0.81 | | | 0.79 |
| 0.88–2.32 | 28 | 1.00 | | 119 | 1.00 | | 100 | 1.00 | |
| 2.33–3.09 | 28 | 0.84 (0.21–3.34) | | 118 | 1.10 (0.60–2.02) | | 123 | 1.01 (0.53–1.94) | |
| 3.10–4.19 | 33 | 1.62 (0.35–7.58) | | 119 | 0.83 (0.42–1.65) | | 114 | 1.28 (0.65–2.55) | |
| 4.20–15.07 | 37 | 0.86 (0.17–4.45) | | 130 | 0.85 (0.41–1.77) | | 99 | 1.35 (0.59–3.08) | |
| PFHxS | | | 0.61 | | | 0.18 | | | 0.28 |
| 0.11–0.50 | 28 | 1.00 | | 117 | 1.00 | | 126 | 1.00 | |
| 0.51–0.68 | 29 | 1.02 (0.27–3.93) | | 132 | 1.04 (0.57–1.90) | | 103 | 0.78 (0.42–1.45) | |
| 0.69–0.93 | 28 | 0.52 (0.12–2.22) | | 106 | 0.71 (0.35–1.42) | | 99 | 0.56 (0.28–1.11) | |
| 0.94–24.92 | 41 | 1.22 (0.27–5.64) | | 131 | 1.39 (0.71–2.73) | | 108 | 0.99 (0.49–1.99) | |
| PFNA | | | 0.33 | | | 0.07 | | | 0.55 |
| 0.08–0.30 | 27 | 1.00 | | 130 | 1.00 | | 124 | 1.00 | |
| 0.31–0.39 | 29 | 0.49 (0.14–1.79) | | 111 | 0.70 (0.40–1.25) | | 124 | 0.79 (0.43–1.42) | |
| 0.40–0.52 | 29 | 0.29 (0.07–1.30) | | 115 | 0.71 (0.40–1.29) | | 111 | 0.70 (0.38–1.30) | |
| 0.53–1.53 | 41 | 0.27 (0.06–1.21) | | 130 | 0.44 (0.24–0.81) | | 77 | 1.03 (0.50–2.13) | |

PFOS: perfluorooctane sulfonate; PFOA: perfluorooctanoic acid; PFHxS: perfluorohexane sulfonate; PFNA: perfluorononanoic acid.

evidence of heterogeneous exposure effects, suggesting that exposure to other risk factors do not make children more susceptible to the prenatal PFAS exposure.

## Strengths and limitations

We used measurements of weight and height recorded by trained nurses as opposed to self-reported data. Measurement errors in weight and height are likely to be non-differential with respect to the exposure, leading to bias towards the null and decreased statistical power. However, the associations between several established risk factors and overweight in this cohort speak against issues with measurement error [19]. Blood sampling was performed in approximately the same gestational week for all women and we thereby reduced the risk of systematic errors in the exposure assessment that might arise in the absence of a standardized sampling period because of pregnancy-related physiological changes. Furthermore, the study was designed to create contrasts between the cases and controls in the risk model thus reducing the risk of misclassification error. The sample of participants can be considered representative for children who are born full term of parents born in Sweden. It should, however, be noted that the proportion of mothers who smoke during pregnancy has declined since the data in the present study was collected.

The study has some limitations. The study participants were from the general population with low levels of exposure and small exposure contrasts. We only had one measurement of ISO-BMI, although BMI is a dynamic state that should preferably be assessed based on repeated measures.

## Conclusions

We did not find any monotonic exposure-response relationships between maternal serum levels of PFOS, PFOA, PFHxS and PFNA in early pregnancy and child overweight at age 4. The absence of an effect was consistent over risk strata and the level of exposure to other risk factors for overweight did consequently not affect children's susceptibility to prenatal PFAS exposure.

## Acknowledgments

The authors thank Åsa Amilon, Agneta Kristensen and Margareta Maxe at the Laboratory of Occupational and Environmental Medicine in Lund for chemical analyses of PFAS in serum and Professor Maria Rosvall for establishing the cohort.

## Author Contributions

**Conceptualization:** Anna Rignell-Hydbom.

**Formal analysis:** Matilda Martinsson, Christel Nielsen.

**Funding acquisition:** Anna Rignell-Hydbom.

**Investigation:** Christel Nielsen.

**Methodology:** Jonas Björk, Lars Rylander.

**Resources:** Ebba Malmqvist, Christian Lindh.

**Software:** Christel Nielsen.

**Visualization:** Matilda Martinsson.

**Writing – original draft:** Matilda Martinsson.

**Writing – review & editing:** Christel Nielsen, Jonas Björk, Lars Rylander, Ebba Malmqvist, Christian Lindh, Anna Rignell-Hydbom.

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
