## [Decision Letter · Decision Letter 0]

12 Dec 2019

PONE-D-19-31602

Intrauterine exposure to perfluorinated compounds and overweight at age 4: A case-control study

PLOS ONE

Dear Dr. Martinsson,

Thank you for submitting your manuscript to PLOS ONE. After careful consideration, we feel that it has merit but does not fully meet PLOS ONE’s publication criteria as it currently stands. Therefore, we invite you to submit a revised version of the manuscript that addresses the points raised during the review process.

We would appreciate receiving your revised manuscript by 01/12/2020. To enhance the reproducibility of your results, we recommend that if applicable you deposit your laboratory protocols in protocols.io, where a protocol can be assigned its own identifier (DOI) such that it can be cited independently in the future. For instructions see: http://journals.plos.org/plosone/s/submission-guidelines#loc-laboratory-protocols

We look forward to receiving your revised manuscript.

Kind regards,

Linglin Xie

Academic Editor

PLOS ONE

Journal Requirements:

1. In ethics statement in the manuscript and in the online submission form, please provide additional information about the patient records used in your retrospective study, in particular regarding children data . Specifically, please ensure that you have discussed whether all data were fully anonymized before you accessed them and/or whether the IRB or ethics committee waived the requirement for informed consent. If patients provided informed written consent to have data from their medical records used in research, please include this information.

2. In your Methods section, please provide additional information about the participant inclusion method and the demographic details of your participants. Please ensure you have provided sufficient details to replicate the analyses such as: a) a description of any inclusion/exclusion criteria that were applied to participant inclusion, b) a table of relevant demographic details, c) a statement as to whether your sample can be considered representative of a larger population.

Reviewers' comments:

Reviewer's Responses to Questions

**Comments to the Author**

1. Is the manuscript technically sound, and do the data support the conclusions?

Reviewer #1: Yes

Reviewer #2: Yes

2. Has the statistical analysis been performed appropriately and rigorously? 

Reviewer #1: Yes

Reviewer #2: Yes

3. Have the authors made all data underlying the findings in their manuscript fully available?

Reviewer #1: Yes

Reviewer #2: No

4. Is the manuscript presented in an intelligible fashion and written in standard English?

Reviewer #1: Yes

Reviewer #2: Yes

5. Review Comments to the Author

Reviewer #1: The paper is generally well written and structured. It basically completes the research goals:

1. investigate the association between maternal serum levels of perfluorooctane sulfonate (PFOS), perfluorooctanoic acid (PFOA), perfluorohexane sulfonate (PFHxS) and perfluorononanoic acid (PFNA) in early pregnancy and overweight in the child at 4 years.

2. assess potential heterogeneity in exposure effect between strata with different levels of other risk factors for overweight.

The authors carefully design the experiment and collect the data. However, in my opinion the paper has some issues regarding to some data analyses and text.

Major comments:

1. Please provide more details (formulations) about calculating the OR between quartiles of maternal serum levels and child overweight at age 4. It is not clear to me how you model the logistic regression.

2. From Table 1, we can see there are obvious differences between the case group and control group in terms of the levels of the risk factors. Could you demonstrate that the differences are negligible or won't affect the final result?

Minor comments:

1. Please give the full name of "OR" when it first appeared.

2. Lines 127 and 128: What do you mean by “All cases (n=416) were included and matched with 2 controls per case (n=832) from the same risk strata, with further matching on sex."? Where were the numbers 416 and 832 derived from and what were they used for?

3. Table 3 and Table 4: Labelling sample size as "Cases" is confusing as it include both "cases" and "controls". "Samples" would be better.

Reviewer #2: The manuscript entitled, "Intrauterine exposure to perfluorinated compounds and overweight at age 4: A case-control study" fills a gap in the current literature on perfluorinated compounds as endocrine disruptors, intrauterine exposure to perfluorinated compounds and association with childhood overweight status during early to mid childhood. All statistical analyses and analytical measurements conducted in the research were described in detail in the manuscript in a sufficient manner to allow for replication by other researchers. The statistical analyses used were also rigorous and appropriate based upon the case-control study design, and there were sufficient explanations given for the matching criteria used and the number of controls per case.

The greatest strengths of the study were the use of serum samples from mothers at the same period of gestation, which allows for more accurate comparisons in the case-control study, and adequate contrasts between cases and controls in the risk model. The greatest limitations of the study were that there were small exposure contrasts within the study population as mentioned in your manuscript and height and weight of children were measured only once for calculation of BMI. I am interested to know if you plan to follow-up with this same study population as the children age to record/monitor their BMI status through puberty and into young adulthood.

Overall, your manuscript is well written and is scientifically sound.

6. PLOS authors have the option to publish the peer review history of their article (what does this mean?). If published, this will include your full peer review and any attached files.

Reviewer #1: No

Reviewer #2: No

---

## [Author Response · Author response to Decision Letter 0]

14 Feb 2020

Response to academic editor and reviewers

Please find our response to each point raised below. The indicated page numbers and lines refer to the marked-up version of the revised manuscript. 

Journal Requirements 

1) In ethics statement in the manuscript and in the online submission form, please provide additional information about the patient records used in your retrospective study, in particular regarding children data. Specifically, please ensure that you have discussed whether all data were fully anonymized before you accessed them and/or whether the IRB or ethics committee waived the requirement for informed consent. If patients provided informed written consent to have data from their medical records used in research, please include this information.

AU: No patient records were used. The children’s data comes from a research questionnaire that their parents agreed to fill out at the regular 4-year health examination, combined with weight and height measured at the same time by the nurse (clarified at p. 4; line 82). 

The ethics committee waived the requirement for informed consent and all analyses were performed on pseudonymized data. This has now been added to the text (p. 5; line 101).

2. In your Methods section, please provide additional information about the participant inclusion method and the demographic details of your participants. Please ensure you have provided sufficient details to replicate the analyses such as: a) a description of any inclusion/exclusion criteria that were applied to participant inclusion, b) a table of relevant demographic details, c) a statement as to whether your sample can be considered representative of a larger population.

AU: Participants were recruited by their pediatric nurse in connection with the routine 4-years health exam. This has now been added to the text (p. 4; line 82). The demographic details of the study sample are presented in Table 1. We have clarified this in the table heading.

 a) Inclusion and exclusion criteria are stated under Study population (p. 4-5).

 b) The demographics that was available is presented in Table 1. The table heading 

 has been clarified.

 c) The sample can be considered representative for children who are born full term 

 of parents born in Sweden. In this context, it should be mentioned that the 

 proportion of women who smoke during pregnancy has declined since the data in 

 the study was collected. This has been added under Strengths and limitations 

 (p. 15; line 317).

3. We note that you have indicated that data from this study are available upon request. PLOS only allows data to be available upon request if there are legal or ethical restrictions on sharing data publicly.

AU: Data contains sensitive personal information and cannot be shared publicly for legal and ethical reasons.

Reviewers' comments

Reviewer #1: 

Major comments:

1. Please provide more details (formulations) about calculating the OR between quartiles of maternal serum levels and child overweight at age 4. It is not clear to me how you model the logistic regression.

AU: In the logistic regressions, where the outcome was overweight at 4 years of age (yes/no), we took two analytical approaches:

i) On the full dataset we included the following variables

- Risk strata

- Difference from strata–specific mean

- Sex 

ii) Within each risk strata (i.e. stratified analysis), we included the following variables

- Difference from strata–specific mean

- Sex 

In both settings, we included a random term for mother to account for the hierarchical nature of the data. 

We have carefully reconsidered each formulation in the text and make the judgement that all the information is there. Although we do respect Reviewer 1’s viewpoint, we would like to refer to Reviewer 2 stating that “All statistical analyses and analytical measurements conducted in the research were described in detail in the manuscript in a sufficient manner to allow for replication by other researchers”.

Please note that when reading the Methods section, we found that a sentence has dropped out of the text. Thus, we have added: “We explored interactions between exposures and risk strata in the full-dataset analyses but the terms were non-significant and therefore not included in the final models.” (p. 8; line 176).

2. From Table 1, we can see there are obvious differences between the case group and control group in terms of the levels of the risk factors. Could you demonstrate that the differences are negligible or won't affect the final result?

AU: Overweight is a multifactorial disease that is affected by several risk factors. Thus, it was expected that the cases would have more present risk factors than the controls. Potential confounding caused by the level of these risk factors was handled by adjusting for the difference in risk score from the strata-specific mean in the final models. However, as in all observational studies, bias due to residual confounding cannot be ruled out. 

Minor comments:

1. Please give the full name of "OR" when it first appeared.

AU: A mistake on our behalf - thanks for pointing it out. Odds ratio has been spelled out both in the abstract and in the main text. We did not use it that many times and hence chose to spell it out rather than abbreviate it.

2. Lines 127 and 128: What do you mean by “All cases (n=416) were included and matched with 2 controls per case (n=832) from the same risk strata, with further matching on sex."? Where were the numbers 416 and 832 derived from and what were they used for?

AU: The 416 children were the overweight children with complete data on covariates in the risk model, and the 832 their matched controls. These were retained for the final analyses. We have combined this paragraph with that above it and added “retained for the final analyses” to clarify (p. 6; line 131).

3. Table 3 and Table 4: Labelling sample size as "Cases" is confusing as it include both "cases" and "controls". "Samples" would be better.

AU: Well spotted! We have changed cases to simply “n” in both tables. 

Reviewer #2: 

I am interested to know if you plan to follow-up with this same study population as the children age to record/monitor their BMI status through puberty and into young adulthood.

AU: No, we are not following this specific population. However, we are planning similar studies in Ronneby municipality in southern Sweden where the population has been heavily exposed to contaminated drinking water after decades of leakage of firefighting foams from an airport into the ground water. We are currently establishing a mother-child cohort of 300 dyads where we collect maternal serum from early pregnancy and onwards, as well as breastmilk and serum from the child. We consider this a more relevant setting because of the small exposure contrasts in the general population. In parallel, register-based studies utilizing growth charts from the child health care in Ronneby are ongoing. The first results are expected towards the end of the year.

---

## [Editor Report · Decision Letter 1]

24 Feb 2020

Intrauterine exposure to perfluorinated compounds and overweight at age 4: A case-control study

PONE-D-19-31602R1

Dear Dr. Nielsen,

We are pleased to inform you that your manuscript has been judged scientifically suitable for publication and will be formally accepted for publication once it complies with all outstanding technical requirements.

With kind regards,

Linglin Xie

Academic Editor

PLOS ONE
---

## [Editor Report · Acceptance letter]

2 Mar 2020

PONE-D-19-31602R1 

Intrauterine exposure to perfluorinated compounds and overweight at age 4: A case-control study 

Dear Dr. Nielsen:

I am pleased to inform you that your manuscript has been deemed suitable for publication in PLOS ONE. Congratulations! Your manuscript is now with our production department. 

With kind regards,

on behalf of

Dr. Linglin Xie 

Academic Editor

PLOS ONE